# Impact of UV Exposure and Incidence of Merkel Cell Carcinoma Between 1990 and 2018 in Austria

**DOI:** 10.3390/cancers17203379

**Published:** 2025-10-20

**Authors:** Boban M. Erovic, Alois Schmalwieser, Rudolf Seemann, Florian Schwabel, Stefan Grasl, Stefan Janik, Matthaeus C. Grasl

**Affiliations:** 1Institute of Head and Neck Diseases, Evangelical Hospital Vienna, Hans-Sachs Gasse 10-12, 1180 Vienna, Austria; rudolf.seemann@gmail.com (R.S.); stefan.janik@meduniwien.ac.at (S.J.); 2Unit of Physiology and Biophysics, University of Veterinary Medicine, Veterinärplatz 1, 1210 Vienna, Austria; alois.schmalwieser@vetmeduni.ac.at (A.S.); florian.schwabel@vetmeduni.ac.at (F.S.); 3Department of Otorhinolaryngology and Head and Neck Surgery, Medical University of Vienna, Währinger Gürtel 18-20, 1090 Vienna, Austria; stefan.grasl@meduniwien.ac.at (S.G.); matthaeus.grasl@meduniwien.ac.at (M.C.G.)

**Keywords:** Epidemiology, Merkel cell carcinoma, UV exposure

## Abstract

**Simple Summary:**

Merkel cell carcinoma (MCC) is a rare but aggressive skin cancer. This study analyzed Austrian cancer registry data between 1990 and 2018 and investigated a possible link between MCC incidence and ultraviolet (UV) radiation. We observed a strong increase in MCC cases, with higher incidence rates in western Austria, where UV radiation levels are also higher. Although MCC is uncommon, its incidence is rising, especially among elderly patients. These findings suggest that cumulative UV exposure contributes to MCC development. Raising awareness among physicians and the public may help to improve early diagnosis and patient outcomes.

**Abstract:**

Background: The purpose of this study was to report on (i) patients’ demographics and (ii) Austrian UV data, and (iii) to examine a possible association between UV exposure and the onset of disease in Austria between 1990 and 2018. Methods: We included all patients diagnosed with MCC and their clinical and demographic data to compare with UV radiation. Results: A total of 538 cases were identified, and the incidence (per 100,000/y) increased from 0.013 to 0.43. The MCC incidence was significantly higher in West Austria (mean incidence 0.269 ± 0.04) compared to East Austria (0.180 ± 0.02 *p* = 0.005). Notably, the mean and maximum UV radiation values were higher in the western (*p* < 0.001) compared to the eastern part of Austria. The sum (*p* = 0.033; r: 0.503) and mean (*p* = 0.019; r: 0.546) UV values correlated significantly with the MCC incidence. Conclusions: The incidence of MCC increased significantly, and higher UV radiation levels in western Austria compared to the east were associated with a correspondingly higher MCC incidence, suggesting a contributing role of UV radiation in general.

## 1. Introduction

Merkel cell carcinoma (MCC) is a rare but highly aggressive neuroendocrine skin cancer that belongs to the group of non-melanoma skin cancers (NMSC) [1,2] with an increasing incidence in several countries worldwide. Large-scale population studies from Scandinavia, the Netherlands, Australia, and the United States have consistently shown a rising trend over the past decades, particularly among elderly individuals between the 6th and 8th decade of life.

Established risk factors include advanced age, immunosuppression, and infection with the Merkel cell polyomavirus (MCPyV). Ultraviolet (UV) radiation has also been implicated in MCC pathogenesis, supported by epidemiological evidence linking higher incidence rates to regions with greater UV exposure [3,4,5,6,7,8,9,10,11,12].

Despite these advances, there remains limited evidence on the long-term association between UV exposure and MCC incidence at a national level in Central Europe [4]. In Austria, where UV radiation varies considerably between alpine western regions and the eastern lowlands, this relationship has not yet been systematically examined. Addressing this knowledge gap is critical for understanding the environmental contribution to MCC and for informing prevention and early detection strategies.

Therefore, we used data from MCC patients who were diagnosed between the years 1990 and 2018. Data were derived from the Austrian National Cancer Registry (ANCR), operated by the National Statistical Institution, Statistics Austria. Furthermore, we gained ultraviolet (UV) data to generate the hypothesis that higher UV radiation levels contribute to an increased incidence of Merkel cell carcinoma (MCC) in Austria. To address this, the present study had three specific objectives: (i) to describe the demographic and clinicopathological characteristics of Austrian MCC patients, (ii) to analyze long-term national UV radiation data, and (iii) to examine the potential relationship between UV exposure and MCC incidence between 1990 and 2018.

## 2. Methods

Anonymous data on all Merkel cell carcinoma (MCC) in patients were derived from the Austrian National Cancer Registry (ANCR). The ANCR is population-based and operated by the National Statistical Institution, Statistics Austria. All new cancer cases in the Austrian resident population are documented in the ANCR. According to the Cancer Statistics Act 1969 and the Cancer Statistics Ordinance 2019, hospitals are obliged to report every case of cancer and every death from cancer. In most federal states, regional or clinical tumor registries act as service providers for the hospitals and carry out data collection and processing in the respective state. Both for data directly from hospitals as well as for data from registries, plausibility and quality criteria apply that are closely linked to international recommendations. For follow-up, and to ascertain death certificate only (DCO) cases, ANCR data is linked with the official causes of death (CoD) statistics derived from Statistics Austria since 1983 (https://www.statistik.at/fileadmin/shared/QM/Standarddokumentationen/B_2/std_b_krebsstatistik.pdf; accessed on 16 October 2025). The study was approved by the Ethics Committee (1892/2017).

In accordance with the requirements for observational studies, we hereby confirm that our manuscript entitled “Impact of UV exposure and incidence of Merkel cell carcinoma between 1990 and 2018 in Austria” has been prepared and reported following the STROBE (Strengthening the Reporting of Observational Studies in Epidemiology) (Appendix A) guidelines.

The attached checklist provides a detailed mapping of each of the 22 STROBE items to the corresponding sections and pages of the manuscript, thereby ensuring transparency, completeness, and adherence to established reporting standards.

Clinicopathological information used for this study were tumor site, histology and behavior, year of diagnosis, sex, age (5 year age groups, up to 95+), staging (based on TNM or description of the extent of disease), diagnosis, region of residence, date of death, and overall survival (ICD-O classification, third edition). Unfortunately, data did not contain information regarding immunosuppression and the MCPyV status. Data on cause of death could not be used since CoD uses ICD-10 coding (ICD-10-GM Version 2025), which does not provide specific codes for MCC. MCC is subsumed under “other malignant neoplasms of the skin”.

The cohort of this study included all Austrian cancer cases with documented MCC diagnoses from 1990 to 2018, and follow-up ended by 31 December 2019 (Data stock from the ANCR on 17 December 2020). Data before 1990 could not be used, as the version of the classification used at that time did not allow the identification of MCC. From 1990 to 2002, tumor site was coded according to ICD-9, and the histology code of ICD-O-1 (International Classification of Diseases for Oncology, Version 1) was used to code the tissue type. Not only were new incoming reports coded according to the new version, but the entire database was converted. The two-digit codes were replaced by corresponding four-digit codes. These are correspondingly unspecific for the period up to 1990, as the previous coding according to the two-digit code led to a considerable loss of information. The codes were re-coded in accordance with international standards. From 2002 to 2006, the ICD-O-2 (International Classification of Diseases for Oncology, Version 2) was used for both localization and histology. The ICD-O-3 (International Classification of Diseases for Oncology, Version 3) has been used since 2006. Since 2020, the ICD-O-3 (International Classification of Diseases for Oncology, Version 3) has been used in revision 2. As with the previous change in coding classification, the entire database was recoded. After going through the whole dataset, registry of the dataset was performed on 31 March 2022.

Using the ICD-O-3 code (C44/8247/3), we identified patients with MCC of the skin documented in the ANCR. Tumor site was determined using the last digit in the ICD-O-3 code for lip (=0), eyelid (=1), ears (=2), other/unspecified parts of the face (=3), scalp and neck (=4), trunk (=5), upper extremities and shoulder (=6), lower extremity and hip (=7), skin, overlapping several sub-areas (=8), and not specified (=9). Codes from 0 to 4 were merged and determined as head and neck. Tumor staging was based on the TNM classification system and available clinical or pathological data. Tumors were categorized as localized, regional, or disseminated disease according to the extent of growth beyond the primary site, lymph node involvement, and the presence of distant metastases. The following codes were used: Tx: primary tumor cannot be assessed; T0: no evidence of tumor; Tis: in situ primary tumor; T1: tumor size less/equal of 2 cm; T2: > 2 but ≤ 5 cm; T3: > 5 cm; T4: primary tumor invades cartilage, fascia, muscle, or bone; Nx: regional lymph nodes could not be assessed; N0: no regional lymph node involvement; N+: regional lymph node metastasis present; Mx: distant metastasis could not be assessed; M0: no evidence of distant metastasis; M+: distant metastasis present. MCC was diagnosed either by an open biopsy or by fine needle aspiration cytology.

Within the ANCR, the municipality code of the patients’ main residence at time of diagnosis is stored and was assigned to federal states by the ANCR for this study. Overall survival was counted from the day of diagnosis until date of death or end of follow-up (31 December 2019). Incidence of disease per 100,000 residents was calculated according to the distribution of the Austrian population from 1990 to 2018 and is indicated as mean ± SEM (standard error of the mean). UV exposure was available for each political district and year (including data of each single day, week, and month) between 1958 and 2001.

The unit of UV irradiation is counted in Watts per square meter (W/m^2^). Subsequently, the UV index is calculated from the highest UV value per day (half-hourly value) by multiplying the numerical value by a factor of 40 and rounding to a whole number. The UV index is therefore a pure numerical value without a unit. The mean (=annual average of all UV values that were measured daily), maximum (=highest UV value of the year), and sum (=the annual sum of all daily measured UV values of a year) of UV values were then calculated for each year according to NUTS (Nomenclature of territorial units for statistics) per county (AT 11 = Burgenland, AT 12 = Lower Austria, AT 13 = Vienna, AT 21 = Carinthia, AT 22 = Styria, AT 31 = Upper Austria, AT 32 = Salzburg, AT 33 = Tyrol, and AT 34 = Vorarlberg) [13]. See Figure 1.

These data were then combined to render nationwide UV data and were correlated with incidence using a Pearson/Spearman’s rho correlation (r). Next, we divided the dataset according to NUTS into East Austria (AT 11, AT 12, and AT 13) and West Austria (AT 21, AT 22, AT 31, AT 32, AT 33, and AT 34).

Patients’ data were analyzed using statistical software R version 3.0.3 (The R Foundation for Statistical Computing, Vienna, Austria) and SPSS software (version 26; IBM SPSS Inc., Armonk, NY, USA). Incidences, as well as UV data (mean and maximal UV values), were separately compared using a paired *t*-test. In particular, paired *t*-tests were applied to compare UV radiation as well as incidences in West compared to East Austria. Given the fact that districts were allocated to either West or East Austria, no subgroup analyses have been performed either for regions or for gender. Fishers’ exact test was used to compare proportions. All tests are two-sided and *p* < 0.05 was considered as statistically significant.

## 3. Results

Between 1 January 1990 and 31 December 2018, a total of 538 patients with MCC of the skin were diagnosed in Austria and documented in the ANCR (Table 1). All clinicopathological and demographic data are shown in Table 2. In general, people diagnosed with MCC were elderly, with 98.7% (*n* = 531) of patients ≥ 50 years and 40.3% (*n* = 217) even ≥80 years. There was a majority of 314 (58.4%) women compared to 224 (41.6%) men, with a 1.4:1 ratio. The median age was 75 years and the median overall survival was 33.9 months.

Diagnosis was carried out for 93.7% (*n* = 504) following surgery, and only for 1.9% (*n* = 10) after biopsy analysis was performed. The method of diagnosis was not retrievable for 24 samples. Although the number of patients with no exactly determined tumor site was 30.1% (*n* = 162) in our cohort, the most common tumor site was the head and neck area, with 37.9% (*n* = 204), followed by the upper (13.6%, *n* = 73) and lower (12.1%, *n* = 65) extremities. The lowest incidence rates were observed of the trunk (6.1%, *n* = 33) and for overlapping regions (0.2%, *n* = 1).

### 3.1. Staging

Staging was available for 308 (57.2%) out of 538 registered cases. Among these, 136 (44.2%) patients presented with regional lymph node metastases (N+), while 44 (14.3%) patients had distant metastatic disease (M+). In 230 cases (42.8%), nodal status or metastatic spread could not be assessed (Nx/Mx) (Table 2).

### 3.2. Incidence

As stated above, 538 cases were identified, and incidence rates were calculated as the number of new MCC cases per 100,000 person-years. The rates were standardized against the national Austrian population of the respective calendar year, but no external standard population was applied (Figure 2). Notably, the incidence of MCC increased from 0.013 to 0.43 and the median incidence of disease per 100,000 was 0.234, with a rising linear trend within the observation period (Figure 2a,b).

The comparison analysis between West and East Austria showed significant differences (Figure 2b) in terms of overall incidence rates. Between 1990 and 2018, MCC incidence was significantly higher in West Austria (mean incidence 0.269 ± 0.04) compared to East Austria (0.180 ± 0.02 *p* = 0.005). Notably, the mean and maximum UV radiation values were higher in the western (*p* < 0.001) compared to the eastern part of Austria. The sum (*p* = 0.033; r: 0.503) and mean (*p* = 0.019; r: 0.546) UV values correlated significantly to the MCC incidence.

Stratifying Austrian regions into “East” and “West” was chosen to reflect the well-documented differences in UV radiation exposure, with western Austria, with its particularly high altitude, alpine areas, receiving significantly higher annual UV levels compared to the predominantly lowland, urbanized areas in the east. Regarding demographic differences, the population size was accounted for when calculating incidence rates; however, population density and age distribution may vary between regions and could act as potential confounders. In particular, Vienna, Austria’s capital, contributes to a higher concentration of elderly residents in the east, while the west is characterized by smaller, more rural communities and strong tourism-related outdoor activity.

### 3.3. UV Data

The maximum, mean, and sum of the UV values in Austria are presented in Table 3 and Figure 3. UV values significantly increased over time, and there is also a difference between the western and eastern parts of Austria.

In particular, the graphical illustration shows a noticeable upward trend for all three UV variables between 1958 and 2001. The comparison analyses showed significant differences in the UV radiation load between West and East Austria. The mean UV radiation load (1581.59 ± 101.76 vs. 1517.84 ± 119.35; *p* < 0.001) was significantly higher in the western part of Austria compared to the eastern part, as well as the maximum UV values (3478.24 ± 234.03 vs. 3347.25 ± 252.60; *p* < 0.001) (Figure 4 and Table 4).

### 3.4. Correlation Analysis Between UV Radiation and Incidence

Overall, the national incidence rate correlated significantly with the mean UV radiation load (*p* = 0.033, r = 0.503) as well as the sum of UV radiation (*p* = 0.033, r = 0.503) countrywide. No significant correlation was detected between incidence and maximal UV radiation (*p* = 0.172, r = 0.337). Moreover, a separate subanalysis for West and East Austria showed no significant correlation in terms of incidence and UV radiation load.

### 3.5. Survival

Univariate and multivariate analyses were performed using Cox-regression. Clinical factors like age, UV exposure, presence of nodal involvement, site of the tumor, and geographic data were included. None of the parameters had an influence on survival, and data on disease-specific survival were not available.

## 4. Discussion

This study presents, for the first time, epidemiological and long-term observational data on Merkel cell carcinoma (MCC) and ultraviolet (UV) radiation in Austria. Our analysis showed that MCC predominantly affects elderly individuals, with the head and neck region and upper extremities as the most common sites of occurrence. These findings are consistent with data from Scandinavia, Finland, the Netherlands, and Australia [14,15,16,17]. In line with previous epidemiological studies, including a large U.S. cohort, we confirmed that female patients are more frequently affected than males [15]. This may, at least in part, be explained by the longer life expectancy of women in Austria [16].

Although tumor size information was unavailable in the Austrian National Cancer Registry (ANCR), the staging data revealed that a considerable proportion of patients presented with advanced disease, with 44% showing nodal involvement and 14% distant metastases among cases with available staging data. Compared with Scandinavian data, where only 10% of patients presented with nodal and 3% with distant metastases, Austrian patients appeared to have more advanced disease at the time of diagnosis [14,15,16,17]. This finding highlights the aggressive clinical course of MCC and the importance of early recognition.

One of the key results of our study is the clear increase in MCC incidence in Austria between 1990 and 2018 (Figure 5), even after adjustment for population changes. This trend mirrors reports from other European countries and the U.S. [14,15,16,17,18,19], underscoring the robustness of our findings. When compared with the most recent study from Sweden, the Austrian incidence rates were substantially higher, ranging from 0.013 to 0.43 per 100,000 person-years, while Swedish rates ranged only from 0.11 to 0.19 per 100,000 person-years [14]. At this point, we do not have data on patients’ geographical distribution within Austrian counties with nodal and metastatic disease, but there is a certain possibility that patients from remote areas wait longer to see a physician compared to patients from urban regions.

The rise in incidence must, however, be interpreted in light of developments in diagnostic practice and awareness. We hypothesize that the introduction of CK20 immunohistochemistry in the 1980s and 1990s and the discovery of the Merkel cell polyomavirus (MCPyV) in 2008 improved diagnostic accuracy and case ascertainment, thereby contributing to the observed increase in incidence [20,21,22]. The growing awareness among physicians likely further amplified detection rates. At the same time, the parallel rise in UV radiation during the same period supports the hypothesis that environmental exposure plays an important role in MCC carcinogenesis [20,21,22,23,24,25,26,27].

Our findings also highlight notable geographic differences within Austria. UV radiation levels were significantly higher in western compared to eastern regions, reflecting alpine topography and higher altitudes. Although mean and cumulative UV values correlated significantly with MCC incidence, no correlation was observed with maximum UV exposure levels. This suggests that the cumulative UV dose, rather than the peak intensity, may be more relevant to MCC pathogenesis. A similar mechanism has been established for basal cell carcinoma, but to our knowledge this has not yet been shown for MCC.

Placing our results in a broader dermatological context, comparable east–west differences have been reported for other UV-related skin cancers. For melanoma, Austrian incidence rates increase with altitude (approximately 2% per 10 m) and are higher in western regions, even after adjustment for urban versus rural differences [28]. Monshi et al. likewise observed the highest melanoma rates in western Austria, with a regional variation in diagnostic practices for thin lesions [29]. Although the data on basal cell carcinoma (BCC) and squamous cell carcinoma (SCC) stratified by Austrian region remain limited, the global rise in BCC incidence underscores UV exposure as a common etiological factor [30]. These parallels strengthen the interpretation that geographic differences in MCC incidence are linked to environmental UV radiation.

Despite these novel insights, our study has several limitations. Tumor size and disease-specific survival data were not available due to incomplete registry information prior to the 2000s, reflecting the limited IT infrastructure during that period. In addition, MCPyV status was not available for Austrian MCC patients diagnosed between 2010 and 2018, although this is likely to become an important diagnostic and therapeutic biomarker in the future. Demographic shifts, particularly population aging, may also have contributed to the rising incidence, as MCC predominantly affects older patients. Improvements in diagnostic awareness, clinical recognition, and histopathological techniques likely influenced case ascertainment and the accuracy of classification. Furthermore, registry completeness remains a limitation, as a large proportion of tumors had an unknown primary site (30%) and staging data were missing in 43% of cases. These limitations underline the need for a cautious interpretation of our results and emphasize the importance of strengthening cancer registry infrastructure for rare malignancies such as MCC. Another limitation of our study is the mismatch in observation periods, as UV radiation data were only available until 2001, whereas MCC incidence was recorded through 2018. Consequently, correlation analyses between UV exposure and MCC incidence reflect only the overlapping period (1990–2001) and do not capture later changes in UV radiation trends. Given that MCC typically develops later in life, a latency period is expected, and the gap between the UV radiation data (ending 2001) and incidence data (extending to 2018) may partly account for this. As UV radiation levels have continued to rise in Austria, extending the UV data to 2018 would likely reinforce rather than alter the observed associations. Nonetheless, this temporal discrepancy may have attenuated correlations and limits conclusions regarding long-term parallel trends between UV radiation and MCC incidence beyond 2001.

## 5. Conclusions

In conclusion, this study shows that MCC is a rare disease in Austria, but the incidence is significantly increasing compared to other countries. This study supports the finding that high UV exposure has a contributing role to an increased incidence of MCC. As a consequence, there is a strong need for further informational campaigns to increase awareness of MCC among clinicians and the general population [31].

## Figures and Tables

**Figure 1 cancers-17-03379-f001:**
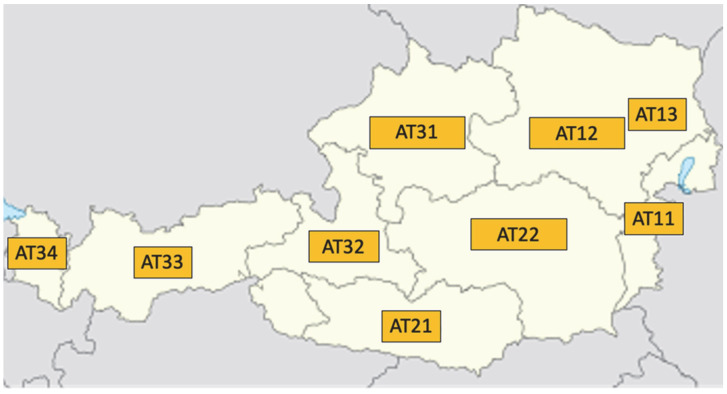
NUTS (Nomenclature of territorial units for statistics) per county (AT 13 = Vienna, AT 11 = Burgenland, AT 12 = Lower Austria, AT 21 = Carinthia, AT 22 = Styria, AT 31 = Upper Austria, AT 32 = Salzburg, AT 33 = Tyrol, and AT 34 = Vorarlberg).

**Figure 2 cancers-17-03379-f002:**
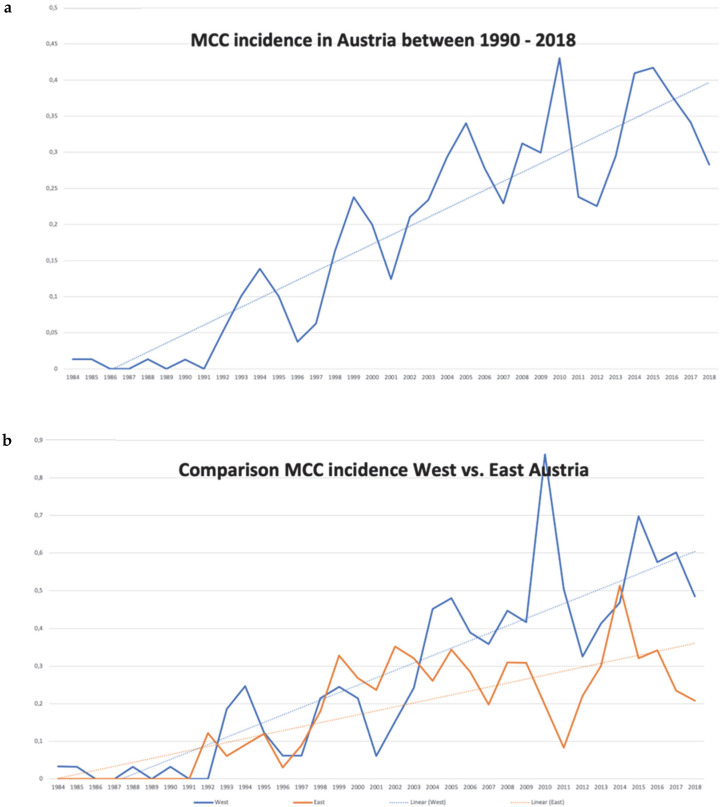
MCC incidence in Austria. Rising incidence of MCC in Austria between 1990 and 2018 (**a**). Note also the incidence of MCC in the western and eastern parts of Austria that from the late 1990s starts to increase in the west (**b**).

**Figure 3 cancers-17-03379-f003:**
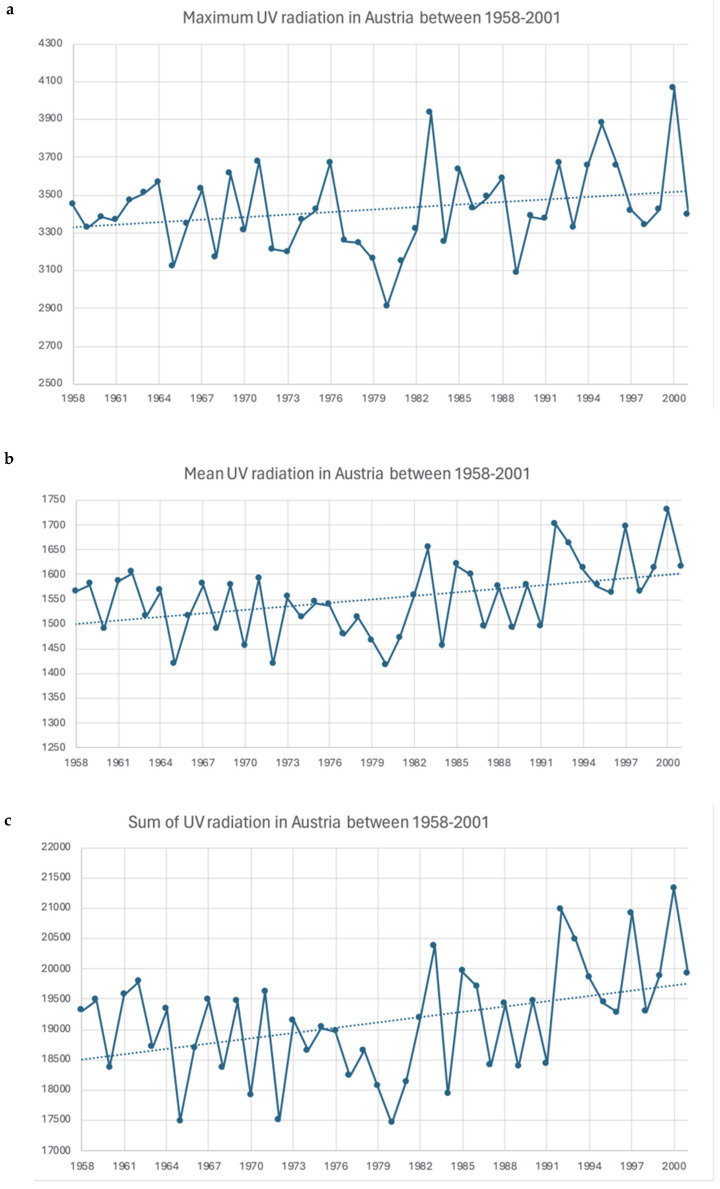
UV radiation data for Austria between 1958 and 2001. Although we indicate maximum (**a**), mean (**b**), and sum (**c**) values of UV irradiation in the figures below, a significant increase of the UV exposure over time can be clearly appreciated.

**Figure 4 cancers-17-03379-f004:**
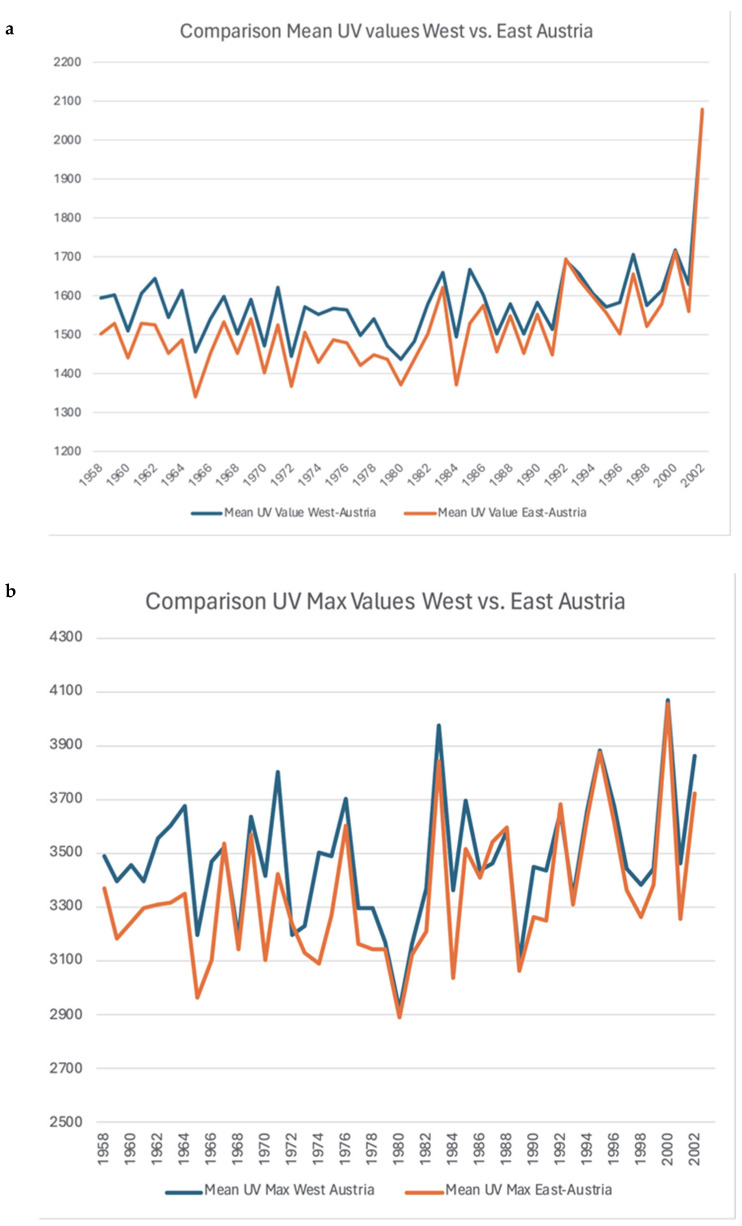
Significant differences in UV radiation between West and East Austria. Mean (**a**) and max (**b**) values of UV radiation in West and East Austria.

**Figure 5 cancers-17-03379-f005:**
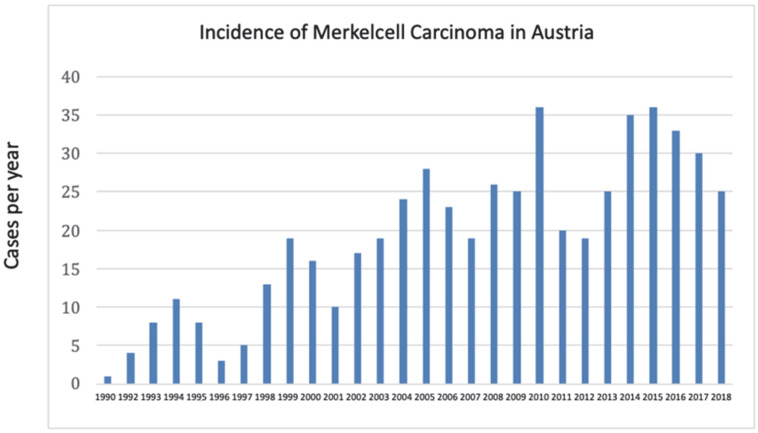
Incidence of Merkel cell carcinoma in Austria.

**Table 1 cancers-17-03379-t001:** Number of patients stratified to the years of diagnosis.

Year of Diagnosis	Number of Patients
1984	1
1985	1
1988	1
1990	1
1992	4
1993	8
1994	11
1995	8
1996	3
1997	5
1998	13
1999	19
2000	16
2001	10
2002	17
2003	19
2004	24
2005	28
2006	23
2007	19
2008	26
2009	25
2010	36
2011	20
2012	19
2013	25
2014	35
2015	36
2016	33
2017	30
2018	25

**Table 2 cancers-17-03379-t002:** Clinicopathological and demographic data of all patients included in this study.

Variables	*n* (%)
Gender	538 (100)
Female	314 (58.4)
Male	224 (41.6)
Age distribution	
Older than 50	531 (98.7)
Older than 80	217 (40.3)
FNA vs. Biopsy	
FNA	10 (1.9)
Biopsy	504 (93.7)
Not known	24 (4.5)
Tumor localization	
Not known	162 (30.1)
Head/Neck	204 (37.9)
Upper extremities	73 (13.6)
Lower extremities	65 (12.1)
Trunk	33 (6.1)
Overlapping regions	1 (0.2)
N-Classification	
Nx	230 (42.8)
N0	172 (55.8)
N+	136 (44.2)
M-classification	
Mx	230 (42.8)
M0	264 (85.7)
M+	44 (14.3)

**Table 3 cancers-17-03379-t003:** UV data from 1958 to 2001 in Austria.

	Annual Mean UV Radiation (J/m^2^)	Highest Monthly UV Radiation of the Year (J/m^2^)
Minimum	1416.49	2909.93
Maximum	1729.40	4062.28
Mean	1552.85	3428.57
Standard Deviation	74.84	228.72
Standard Error of Mean	11.28	34.48

**Table 4 cancers-17-03379-t004:** UV data (J/m^2^) from 1958 to 2001 in western and eastern Austria.

		Western (J/m^2^)	Eastern (J/m^2^)	*p*-Value
Maximum		3478.25	3347.25	<0.001
	Standard Deviation	234.03	252.61	
	Standard Error	34.89	37.66	
	95% CI	3409.9–3546.6	3273.4–3421.1	
Mean		1581.60	1517.84	<0.001
	Standard Deviation	101.77	119.36	
	Standard Error	15.17	17.79	
	95% CI	1551.9–1611.3	1483.0–1552.7	

## Data Availability

The datasets generated and/or analyzed during the current study are available from the corresponding author on reasonable request.

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
