# Peer review of "Impact of UV Exposure and Incidence of Merkel Cell Carcinoma Between 1990 and 2018 in Austria"

_cancers, 2025, doi:10.3390/cancers17203379_

Round 1

Reviewer 1 Report

Comments and Suggestions for Authors

Dear Authors,

Thank you for the oppotunity of reviewing your interesting manuscript. The study addresses an important and timely topic and presents valuable national data. However, before it is suitable for publication it has major concerns. The authors should address issues of clarity, causal interpretation, and methodological transparency, especially concerning the mismatch between UV data and cancer incidence periods, and provide stronger justification for geographic grouping and statistical choices. Please, find below my comments.

ABSTRACT:

  1. The sentence structure is sometimes awkward (“to describe i) patients ́demographic, ii) Austrian UV-data and iii) to analyse…”). A minor rephrasing to improve clarity and grammar is recommended.
  2. The conclusion slightly overstates causality (“resulted in a higher MCC incidence could be determined”). Please, consider using more cautious language (e.g., “was associated with” rather than “resulted in”).

INTRODUCTION:

  1. A clearer statement of the study’s primary hypothesis and specific objectives would strengthen the framing.
  2. A clearer background of the state of the art could help the readers to understand the importance of this work.

METHODS:

  1. A “study design” section should be added. Also, the STROBE statement should be followed. Please, provide the STROBE as supplemental material.
  2. It would help to clarify the period mismatch: UV radiation data end in 2001, whereas MCC incidence extends to 2018. The implications of this discrepancy for correlation analyses should be discussed more explicitly.
  3. Please justify the decision to group Austrian regions into “East” and “West” and clarify whether population density and demographic differences were adjusted for.
  4. Provide more detail on how incidence rates were age-standardised and which standard population (if any) was used.
  5. Indicate the exact years of Ethics Committee approval and registry data freeze more clearly.
  6. The description of staging (e.g., use of “Nx”, “Mx”) could be clarified, and typographical inconsistencies (e.g., “patients ś main residence”) should be corrected.
  7. It is essential that the authors provide a detailed description of the analyses (incidence, UV radiation) included in the manuscript. This is essential for the transparency of the study. It is also essential that the subgroup analyses (gender, regions…) be detailed. Major issue

RESULTS:

  1. Major issue: Enhance figure captions and provide confidence intervals where relevant. Figure 3 and 4 are difficult to visualize and read, data should be provided to improve understanding.
  2. Major issue: A table with all the years results should be added, please.
  3. A large proportion of tumours with unknown site (30%) and missing staging data (about 43%) is a limitation that should be highlighted more explicitly.
  4. Please, report confidence intervals for key incidence estimates.
  5. Correlation coefficients are given, but the strength of association should be interpreted cautiously, and any correction for multiple testing should be stated.

DISCUSSION:

  1. Speculation about diagnostic awareness (CK19, MCPyV) is interesting but could be better supported with references or rephrased as hypotheses.
  2. Some sentences are long and would benefit from restructuring for readability.
  3. The limitations section is appropriate but could be expanded to include possible confounding factors (e.g., population ageing, changes in reporting practices).

CONCLUSIONS:

  1. The conclusions are broadly supported by the data but again somewhat strong in implying causation.
  2. A slightly more balanced closing statement would be advisable (e.g., “Our findings support an association between higher cumulative UV exposure and MCC incidence” rather than “high UV exposure is linked to an increased incidence”).

Author Response

Reviewer #1:

Dear Authors,

Thank you for the opportunity of reviewing your interesting manuscript. The study addresses an important and timely topic and presents valuable national data. However, before it is suitable for publication it has major concerns. The authors should address issues of clarity, causal interpretation, and methodological transparency, especially concerning the mismatch between UV data and cancer incidence periods, and provide stronger justification for geographic grouping and statistical choices. Please, find below my comments.

ABSTRACT:

  1. The sentence structure is sometimes awkward (“to describe i) patients ́demographic, ii) Austrian UV-data and iii) to analyse…”). A minor rephrasing to improve clarity and grammar is recommended.

Answer: Sentence was rephrased (page 2). All changes are marked yellow in the text.

  1. The conclusion slightly overstates causality (“resulted in a higher MCC incidence could be determined”). Please, consider using more cautious language (e.g., “was associated with” rather than “resulted in”).

Answer: The conclusion was rephrased (page 2). All changes are marked yellow in the text.

INTRODUCTION:

  1. A clearer statement of the study’s primary hypothesis and specific objectives would strengthen the framing.

Answer: Introduction regarding hypothesis and objectives was modified (page 3). All changes are marked yellow.

  1. A clearer background of the state of the art could help the readers to understand the importance of this work.

Answer: Introduction was completely modified (page 3). All changes are marked yellow.

METHODS:

  1. A “study design” section should be added. Also, the STROBE statement should be followed. Please, provide the STROBE as supplemental material.

Answer: thank you very much! Study design is now included in the Materials and Methods section (page 4) and STROBE statement was followed and is now available as supplemental material (page 22). All changes are marked yellow.

  1. It would help to clarify the period mismatch: UV radiation data end in 2001, whereas MCC incidence extends to 2018. The implications of this discrepancy for correlation analyses should be discussed more explicitly.

Answer: Thank you for this comment. A limitation of our analysis is the mismatch in observation periods, as UV radiation data were only available until 2001, whereas MCC incidence was assessed through 2018. Consequently, correlation analyses between UV exposure and MCC incidence reflect only the overlapping period from 1990 to 2001 and do not capture subsequent changes in UV radiation trends. Moreover, due to the appearance of MCC relatively late in life, a certain latency period can be expected. The time gap between the end of the times series of UV radiation (2001) and of incident rate (2018) reflects this latency period in our analysis. The increase in UV radiation is still ongoing and including UV data up to 2018 in analysis would therefore not alter results. Nevertheless, we have to confirm that this temporal discrepancy may have attenuated observed associations and limits the ability to draw conclusions regarding long-term parallel trends between UV radiation and MCC incidence beyond 2001. All changes are marked yellow (Page 10).

  1. Please justify the decision to group Austrian regions into “East” and “West” and clarify whether population density and demographic differences were adjusted for.

Answer: We thank the reviewer for this important comment. The grouping of Austrian regions into “East” and “West” was chosen to reflect well-documented differences in UV radiation exposure, with western Austria, particularly high altitude, alpine areas, receiving significantly higher annual UV levels compared to predominantly lowland, urbanized areas in the east. Regarding demographic differences, we accounted for population size when calculating incidence rates; however, we acknowledge that factors such as population density and age distribution may vary between regions and could act as potential confounders. In particular Vienna, Austria´s capital, contributes to a higher concentration of elderly residents in the east, while the west is characterized by smaller, more rural communities and strong tourism-related outdoor activity (see page 6+7).

We have clarified this rationale in the revised manuscript and explicitly highlighted the lack of adjustment for demographic differences as a limitation. These differences may act as potential confounders and were therefore acknowledged as limitations in our analysis (page 7+8). All changes are marked yellow.

  1. Provide more detail on how incidence rates were age-standardised and which standard population (if any) was used.

Answer: Incidence rates were calculated as the number of new MCC cases per 100.000 person-years. The rates were standardized against the national Austrian population of the respective calendar year. No external standard population was applied (page 7). All changes are marked yellow.

  1. Indicate the exact years of Ethics Committee approval and registry data freeze more clearly.

Answer: The Ethics Committee approval was received in 2022 (see page 11) and registry data freeze was done on march 31st, 2022 (see page 5). Information was included in the manuscript and marked yellow.

  1. The description of staging (e.g., use of “Nx”, “Mx”) could be clarified, and typographical inconsistencies (e.g., “patients ś main residence”) should be corrected.

Answer: TNM classification is represented in detail (page 5 + 7) and typographical errors were corrected. All changes are marked yellow.

  1. It is essential that the authors provide a detailed description of the analyses (incidence, UV radiation) included in the manuscript. This is essential for the transparency of the study. It is also essential that the subgroup analyses (gender, regions…) be detailed.

Answer: Paired t-tests were applied to compare UV radiation as well as incidences of west compared to east Austria. Given the fact that districts were allocated to either West or East Austria, no subgroup analyses have been performed for regions or gender either. All changes are marked yellow (page 6).

RESULTS:

  1. Major issue: Enhance figure captions and provide confidence intervals where relevant. Figure 3 and 4 are difficult to visualize and read, data should be provided to improve understanding.

Answer: We absolutely agree with Reviewer 1 that the current version of Figure 3 and 4 are difficult to read. Therefore, all figures are provided and uploaded as separate files, which should ease visualization and interpretation. New figures were inserted into the manuscript and figures legends are marked yellow (page 17 - 20).

  1. Major issue: A table with all the years results should be added, please.

Answer: a new table, named as Table 1, was added that shows the year of diagnosis and the number of patients that has been diagnosed (page 2). All changes are marked yellow.

  1. A large proportion of tumours with unknown site (30%) and missing staging data (about 43%) is a limitation that should be highlighted more explicitly.

Answer: Missing data as a limitation were explicitly mentioned in the ´limitation section´ of the discussion (page 10). All changes are marked yellow.

  1. Please, report confidence intervals for key incidence estimates.

Answer: With regards to confidence intervals, as requested, we added confidence intervals in Table 3 (page 14). All changes are marked yellow.

  1. Correlation coefficients are given, but the strength of association should be interpreted cautiously, and any correction for multiple testing should be stated.

Answer: we absolutely agree, and we modified the discussion as stated above (page 10). We did not perform multiple testing.

DISCUSSION:

  1. Speculation about diagnostic awareness (CK20, MCPyV) is interesting but could be better supported with references

Answer: References on this specific topic were included for better support for diagnostic awareness: Tetzlaff MT, Nagarajan P. Update on Merkel Cell Carcinoma. Head Neck Pathol. 2018;12:31–43.

Yeni Erdem B, Baykal C, Ozluk Y, Ahmed MA, Kozanoglu E, Saip P, Buyukbabani N, Ozturk Sari S. Evaluating CK20 and MCPyV Antibody Clones in Diagnosing Merkel Cell Carcinoma. Endocr Pathol. 2025;36:1. All changes are marked yellow (page 10+25).

  1. Some sentences are long and would benefit from restructuring for readability.

Answer: Discussion was adopted for better readability (pages 9 and 10). All changes are marked yellow.

  1. The limitations section is appropriate but could be expanded to include possible confounding factors (e.g., population ageing, changes in reporting practices).

Answer: The limitation section was rephrased (page 10). All changes are marked yellow in the text.

CONCLUSIONS:

  1. The conclusions are broadly supported by the data but again somewhat strong in implying causation.

Answer: The conclusions were rephrased (page 11). All changes are marked yellow in the text.

  1. A slightly more balanced closing statement would be advisable (e.g., “Our findings support an association between higher cumulative UV exposure and MCC incidence” rather than “high UV exposure is linked to an increased incidence”).

Answer: The closing statement was rephrased (page 11). All changes are marked yellow in the text.

Reviewer 2 Report

Comments and Suggestions for Authors

The Authors should be commended for producing a well conducted and presented study.

A few suggestions:

1 Tumor location: Unknown 30.1%, U limb 13.6%, L limb 12.1% and Trunk 6.1%. Given the considerable larger surface area on the trunk these locations strengthen the association between MCC and UV exposure. An extension of the strength of this association would be the comparison between (a) Shoulder to elbow with below the elbow AND (b) thigh compared to below the knee. It would be expected that the higher incidence of UV on the distal parts of the limbs (from wearing short sleeve shirts and shorts in summer) would show a even stronger UV association. 

2 Minor formating: insert a space on line 163 between the word between and 1990, line 228 between Merkel and the word cell, line 246 between the words basal and cell.   

Author Response

Reviewer #2:

  1. Tumor location: Unknown 30.1%, U limb 13.6%, L limb 12.1% and Trunk 6.1%. Given the considerable larger surface area on the trunk these locations strengthen the association between MCC and UV exposure. An extension of the strength of this association would be the comparison between (a) Shoulder to elbow with below the elbow AND (b) thigh compared to below the knee. It would be expected that the higher incidence of UV on the distal parts of the limbs (from wearing short sleeve shirts and shorts in summer) would show a even stronger UV association.

Answer: This is a fantastic idea and suggestion, but unfortunately, we do not have data for further statistical analysis.

  1. Minor formatting: insert a space on line 163 between the word between and 1990, line 228 between Merkel and the word cell, line 246 between the words basal and cell.

Answer: Formatting was performed. All changes are marked yellow.

Reviewer 3 Report

Comments and Suggestions for Authors

This paper is dealing with UV exposure and incidence of Merkel cell carcinoma (MCC) in Austria. The authors provide data on MCC stages (local tumor, with lymph node metastases, with distant metastases), with a total of 538 cases, from the Austrian National Cancer Registry (ANCR) from 1990 to 2018 and correlate these data to the UV exposure in three districts in East-Austria and six districts in West-Austria. The UV data mean, maximum, and sum were evaluated between 1958 and 2001 which were higher in West Austria.

The result shows an increasing incidence of MCC in Austria between 1990 and 2018 and moreover the incidence was higher in West-Austria where the UV exposure was higher.

The study presents interesting epidemiological observations on MCC but there are some points of criticism:

  1. There is an Austrian district mentioned AT13 (l. 123) but not included in the map of Fig. 1. Has AT9 (Vienna) not been considered? What are the main environmental arguments to separate Austria into these parts (e.g. to assign Styria = AT22 to West-Austria)?
  2. Why have UV data only up to 2001 been assessed? For MCC arising near 2018 UV radiation after 2001 might be relevant. This point should be commented on in the Discussion.
  3. A MCC incidence in Austria of up to 3.45 is stated (l. 25, l. 215). This high value appears strange (even in Australia it is not higher than 2.5) and it is also not reflected in Fig. 2.
  4. In the Discussion the authors mention that most carcinomas were high stage tumors with presence of nodal and even distant metastatic disease (l. 203f.). However, Table 1 notes only 44% nodal metastasis and 14% distant metastasis for the cases with known stage.
  5. 223, l. 230: CK19 is not a specific marker for MCC as mentioned by the authors but rather a marker for normal Merkel cells. The typical marker for MCC is CK20 (for early literature according to PubMed see Moll et al. 1992 Am J Pathol, Miettinen 1995 Hum Pathol). The statements on diagnostic markers in the literature and the increasing incidence of MCC should be modified accordingly.
  6. The authors should comment on whether similar differences in incidence between Western and Eastern Austria have been observed concerning other UV induced skin cancers such as melanoma, basal cell carcinoma or squamous cell carcinoma.
  7. In the graphs shown the labeling of values and years is extremely small and barely legible.

Author Response

Reviewer #3

This paper is dealing with UV exposure and incidence of Merkel cell carcinoma (MCC) in Austria. The authors provide data on MCC stages (local tumor, with lymph node metastases, with distant metastases), with a total of 538 cases, from the Austrian National Cancer Registry (ANCR) from 1990 to 2018 and correlate these data to the UV exposure in three districts in East-Austria and six districts in West-Austria. The UV data mean, maximum, and sum were evaluated between 1958 and 2001 which were higher in West Austria.

The result shows an increasing incidence of MCC in Austria between 1990 and 2018 and moreover the incidence was higher in West-Austria where the UV exposure was higher.

The study presents interesting epidemiological observations on MCC but there are some points of criticism:

  1. There is an Austrian district mentioned AT13 (l. 123) but not included in the map of Fig. 1. Has AT9 (Vienna) not been considered?

Answer: Sorry, this was typo. Vienna is not AT9 but AT13 and it has been considered in statistical analysis. Figure 1 was modified correctly (page 15).

  1. What are the main environmental arguments to separate Austria into these parts (e.g. to assign Styria = AT22 to West-Austria)?

Answer: We thank the reviewer for this important comment. The decision to group Austrian regions into East (AT11–13) and West (AT21–34) was based on environmental and geographical considerations, particularly UV radiation exposure. Western Austria is characterized by alpine and high-altitude regions (Carinthia, Styria, Upper Austria, Salzburg, Tyrol, and Vorarlberg), where thinner atmosphere, snow reflectance, and clearer skies result in higher mean and maximum UV values. These regions also have a strong tourism-related outdoor activity profile, which further increases cumulative UV exposure. In contrast, eastern Austria (Vienna, Lower Austria, and Burgenland) consists mainly of lowlands and urbanized areas, with lower altitudes, denser populations, and lower UV radiation levels. For these reasons, Styria (AT22) and other alpine federal states were assigned to West-Austria (pages 7-9).

  1. Why have UV data only up to 2001 been assessed? For MCC arising near 2018 UV radiation after 2001 might be relevant. This point should be commented on in the Discussion.

Answer: Thank you very much for this comment. A limitation of our analysis is the mismatch in observation periods, as UV radiation data were only available until 2001, whereas MCC incidence was assessed through 2018. Consequently, correlation analyses between UV exposure and MCC incidence reflect only the overlapping period from 1990 to 2001 and do not capture subsequent changes in UV radiation trends. Moreover, due to the appearance of MCC relatively late in life, a certain latency period can be expected. The time gap between the end of the times series of UV radiation (2001) and of incident rate (2018) reflects this latency period in our analysis. The increase in UV radiation is still ongoing and including UV data up to 2018 in analysis would therefore not alter results. Nevertheless, we have to confirm that this temporal discrepancy may have attenuated observed associations and limits the ability to draw conclusions regarding long-term parallel trends between UV radiation and MCC incidence beyond 2001. All changes are marked yellow (Page 10).

  1. A MCC incidence in Austria of up to 3.45 is stated (l. 25, l. 215). This high value appears strange (even in Australia it is not higher than 2.5) and it is also not reflected in Fig. 2.

Answer: Sorry, this was a terrible typo. We went through all figures and data and modified the abstract, results and discussion (pages 2, 7 + 9). All changes are marked yellow.

  1. In the Discussion the authors mention that most carcinomas were high stage tumors with presence of nodal and even distant metastatic disease (l. 203f.). However, Table 1 notes only 44% nodal metastasis and 14% distant metastasis for the cases with known stage.

Answer: Yes, you are right and we modified the discussion regarding this issue (page10). All changes are marked yellow.

  1. 223, l. 230: CK19 is not a specific marker for MCC as mentioned by the authors but rather a marker for normal Merkel cells. The typical marker for MCC is CK20 (for early literature according to PubMed see Moll et al. 1992 Am J Pathol, Miettinen 1995 Hum Pathol). The statements on diagnostic markers in the literature and the increasing incidence of MCC should be modified accordingly.

Answer: Sorry, this was a terrible and stupid error! Of course, it is CK230 and NOT CK19! Text is now modified accordingly to CK20 (page 9). All changes are marked yellow.

  1. The authors should comment on whether similar differences in incidence between Western and Eastern Austria have been observed concerning other UV induced skin cancers such as melanoma, basal cell carcinoma or squamous cell carcinoma.

Answer: Thank you very much for this excellent comment. We went through the literature and found a couple of studies on this topic. References are included and the discussion was modified accordingly (page 10). All changes were marked yellow.

  1. In the graphs shown the labeling of values and years is extremely small and barely legible.

Answer: We absolutely agree that the current version of the graphs are difficult to read. Therefore, all figures are provided and uploaded as separate files, which should ease visualization and interpretation (page 15-20).

Round 2

Reviewer 3 Report

Comments and Suggestions for Authors

The authors revised the paper according to several of the points of criticism raised. However, there are still some points still to be considered: 
1. Fig 1: Vienna AT 13 should also be labeled on the map.
               AT 9 is given on the map. What territorial unit is represented?
2. Fig. 2: The labels a, b are missing in the figure.
3. Fig. 3: The labels a, b, c are missing in the figure.
4. Reference no. 13 remained unchanged. This paper is not valid for MCC because it is dealing with Merkel cells exclusively. Two papers dealing with keratin 20 as a marker for MCC were suggested in the initial review.

Author Response

Dear Editor!

The reference list has been increased to 31 citations and all references were modified regarding et. al as suggested.

Reviewer #3

  1. Fig 1: Vienna AT 13 should also be labeled on the map. AT 9 is given on the map. What territorial unit is represented?

Answer: The label AT13 represents the territorial unit of Vienna. AT9 is deleted on the map.

  1. 2: The labels a, b are missing in the figure.

Answer: Figures are labeled.

3. Fig. 3: The labels a, b, c are missing in the figure.

Answer: Figures are labeled.

  1. Reference no. 13 remained unchanged. This paper is not valid for MCC because it is dealing with Merkel cells exclusively. Two papers dealing with keratin 20 as a marker for MCC were suggested in the initial review.

Answer: New references are now included as suggested.